# The actin regulator profilin 1 is functionally associated with the mammalian centrosome

Michaela Nejedlá[1,*], Anastasiya Klebanovych[2,*], Vadym Sulimenko[2], Tetyana Sulimenko[2], Eduarda Dráberová[2], Pavel Dráber[2], Roger Karlsson[1]

**Profilin 1 is a crucial actin regulator, interacting with monomeric actin and several actin-binding proteins controlling actin polymerization. Recently, it has become evident that this profilin isoform associates with microtubules via formins and interferes with microtubule elongation at the cell periphery. Recruitment of microtubule-associated profilin upon extensive actin polymerizations, for example, at the cell edge, enhances microtubule growth, indicating that profilin contributes to the coordination of actin and microtubule organization. Here, we provide further evidence for the profilin-microtubule connection by demonstrating that it also functions in centrosomes where it impacts on microtubule nucleation.**

## Introduction

The centrosome is the major microtubule-organizing structure in eukaryotic cells and as such crucial for intracellular architecture, cell polarity, and directional migration; consequently it has intrigued scientists for decades (1). Studies of the complex protein composition of mammalian centrosomes (2, 3, 4) combined with high-resolution imaging of centrosome behavior in cultured cells have led to the realization that in addition to being intimately linked with the microtubule system they are also closely connected to the actin microfilament system (2, 5, 6, 7, 8, 9, 10), reflecting the tight coordination and polarization of microtubule and actin organization (11, 12, 13). Several actin regulatory components such as WASH, Arp2/3, cofilin, and members of the formin family (2, 7, 14) have been found to be associated with the centrosome and likely to govern centrosome-linked actin reorganization. Examples of such activities are for instance the alteration of actin organization during synapse formation in immune cells with subsequent centrosome relocation and changes in microtubule distribution (9) and the centrosomal actin polymerization linked to

spindle formation during mitosis in somatic mouse and human cells (15). Moreover, centrosomal proteins also have been reported to promote actin polymerization-driven cell protrusions at the cell cortex in cancer cells (16).

The γ-tubulin ring complex (γTuRC) is crucial to centrosome-derived microtubule nucleation (17). It consists of 14 γ-tubulin (γ-Tb) molecules (18) arranged together with γ-tubulin complex proteins (GCP) 2-6 (4, 19) into a cone-shaped structure designed to nucleate microtubules (20, 21). In addition, recent cryo-electron microscopy revealed that the lumen of γTuRC contains an actin-like molecule associated with both the GCPs and γ-Tbs (22, 23, 24), which appears to be important for γTuRC-dependent microtubule nucleation (23). Although, currently unclear whether it is β- or γ-actin, the observation unveils a direct association between key components of the γTuRC and the microfilament system. The connection between actin and γ-Tb is further emphasized by reports of the latter localizing to the actin-rich cell periphery and influencing stress fiber formation (25).

Profilin is a principal control component of actin polymerization; it brings polymerization-competent, ATP-bound actin monomers to sites where polymerization is called for by activated actin nucleation and elongation factors such as members of the Ena/Vasp, formins, WASP, and WAVE families of actin binding proteins whose interaction with profilin is established via poly(L-proline)-sequences, reviewed by references 26, 27, 28, and 29.

Here, we have continued our studies of the interplay between actin and microtubules which were initiated by the identification of profilin as a control component of microtubule elongation at the cell periphery (30), and report that regulation of microtubules by profilin also extends to the centrosome. It is found that its occurrence in the centrosome impacts on centrosomal accumulation of γ-Tb and microtubule nucleation. Unless otherwise stated, "profilin" refers to profilin 1 throughout this text.

[1]Department of Molecular Biosciences, The Wenner-Gren Institute, Stockholm University, Stockholm, Sweden   [2]Department of Biology of Cytoskeleton, Institute of Molecular Genetics of the Czech Academy of Sciences, Prague, Czech Republic

Correspondence: roger.karlsson@su.se; paveldra@img.cas.cz
Michaela Nejedlá's present address is Institute of Botany, Czech Academy of Sciences, Průhonice, Czech Republic
*Michaela Nejedlá and Anastasiya Klebanovych contributed equally to this work

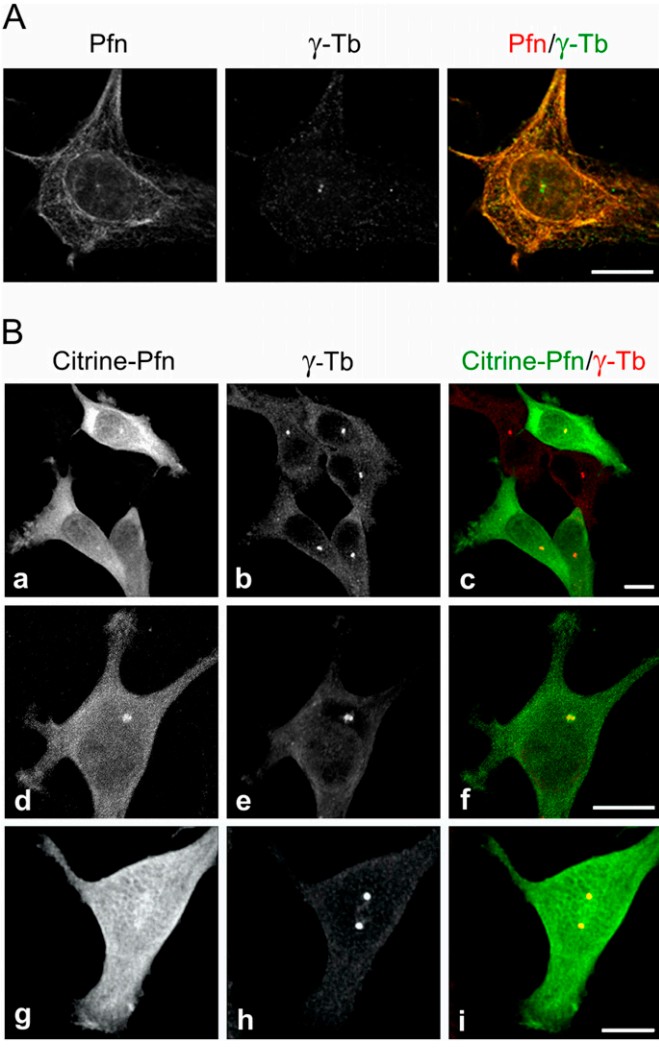

**Figure 1. Profilin is present in centrosomes throughout the cell cycle.**
Fluorescence microscopy revealed the co-localization of profilin and γ-tubulin. **(A)** Ab staining of profilin and γ-Tb in B16 cells after a brief pretreatment with 0.1% Triton X-100 before fixation (see text). High-resolution confocal microscopy (AiryScan). **(B)** B16 cells expressing citrine-profilin (a, d, g; green) were captured at different cell cycle stages. Cells were permeabilized with 10 $\mu$M digitonin, fixed, and stained for γ-tubulin (b, e, h; red). Superposition of images in (c, f, i). The cell in (d, e, f) was captured at an early G2-stage of the cell cycle and the one in (g, h, i) at late G2/prophase. Micrographs were obtained by spinning disk microscopy. Scale bars (A) and (B): 10 $\mu$m.

## Results

To learn more about the microtubule association of profilin in the perinuclear region and its possible association with the centrosome we applied a brief washout in accordance with our previous study (30) to reduce accumulation of profilin in this region and unveil possible distinct features of its distribution by immunofluorescence microscopy of mouse melanoma B16 cells. Strikingly enough, this approach enabled visualization of a prominent fluorescence in one or two discrete spots of size, character and position suggesting they represented centrosomes. Co-labeling of γ-tubulin verified that this

indeed was the case (Fig 1A) in agreement with earlier mass spectroscopy data identifying profilin as a component of isolated centrosomes (2). Moreover, the fluorescence intensity emanating from the centrosome after profilin Ab staining increased 1.6 times in cells during mitosis (Fig S1A–C).

To further investigate the centrosomal distribution of profilin and to exclude the risk that the Ab used either cross-reacted with profilin 2, whose expression is up-regulated in B16 profilin 1 knockout cells (KO27) (31) or contained autoantibodies recognizing the centrosome (32), a fluorescent variant of profilin with citrine fused intra-molecularly was expressed in B16 cells. This chimeric citrine-profilin molecule binds actin and poly(L-proline) (31); the two principal interaction properties of profilin (27, 33, 34). After digitonin extraction, fixation, labeling of γ-tubulin with Ab and microscopy, the distribution of citrine-profilin to centrosomes as identified by the γ-tubulin staining was obvious, and a fine reticular profilin fluorescence juxtaposed to centrosomes was often particularly prominent (Fig 1B). Altogether, this demonstrates that profilin associates with mammalian centrosomes.

Based on these microscopy results and our previous observations of profilin operating as a regulator of microtubule elongation (30), we decided to test if profilin interacts with the γ-tubulin ring complex (γTuRC), which is essential for nucleation of microtubules from centrosomes. Therefore, a series of immunoprecipitation (IP)-experiments of extracts of B16 cells was performed using Abs to profilin, γ-tubulin (γ-Tb), and γ-Tb complex protein (GCP)-2 (Fig 2A–C). The material precipitated with the anti-profilin Ab was then found to contain γ-Tb, GCP2 and GCP4, and reciprocal IPs of γ-Tb and GCP2, respectively, revealed co-precipitation of profilin, indicating interaction of profilin with γTuRC. To further validate the obtained data, we performed IP experiments using extracts of B16 wild-type and KO27 cells. Congruent with a profilin-γTuRC interaction, the profilin Abs co-precipitated γ-tubulin, GCP2, GCP4 and actin from wild-type B16 cells but not from the profilin 1-lacking KO27 (Fig 2D). Moreover, the profilin-γTuRC association was not limited to B16 cells, as similar observations also were made with extracts of human colorectal adenocarcinoma Caco-2 cells (Fig S2A and B). Isotype controls for the IP experiments are shown in (Fig S2C and D). Together these results provide strong evidence for profilin being a partner to one or more components of γTuRC. To test whether the association of profilin to γTuRC was due to a direct interaction with γ-tubulin, different GST-fusions of γ-tubulin were used in pull-down experiments. However, an interaction could not be established (data not shown), suggesting that the association of profilin with γTuRC requires one or more additional components.

Next we again took advantage of the KO27 cells, to determine de novo formation of microtubules from interphase centrosomes. To that end, nocodazole-washout experiments were performed as described previously (35, 36). As demonstrated by immunofluorescence microscopy of α-tubulin after drug removal and 2 min of recovery in fresh medium at 37°C, the microtubule array reformed more rapidly in KO27 than in the profilin-expressing control cells (Fig 3A). Quantification of the α-tubulin immunofluorescence revealed approximately three times more intensive signal in the profilin-depleted cells than in the control (Fig 3B), and, interestingly, a similar quantification of γ-tubulin immunofluorescence unveiled that also γ-tubulin accumulates above wild-type level at

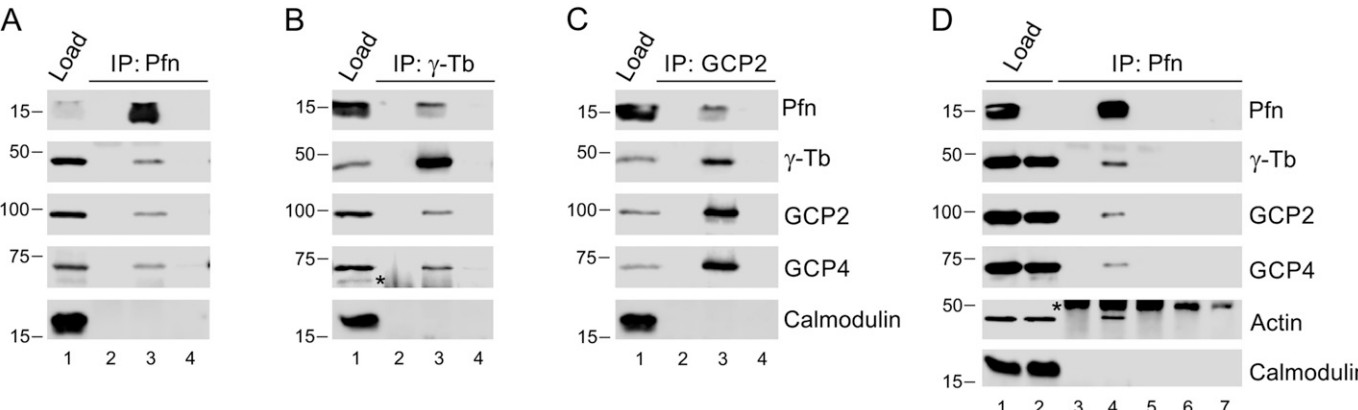

**Figure 2. Profilin interacts with γ-TuRC.**
**(A, B, C)** Extracts of B16 cells were precipitated with immobilized Abs to profilin (A), γ-tubulin (B) or GCP2 (C), followed by Western blotting and probing with Abs against profilin (Pfn), γ-tubulin (γ-Tb), GCP2, GCP4, and calmodulin (negative control) as indicated (C). In each blot (A, B, C), the sample lanes denoted 1–4 show: total cell extract, gel load (1); no cell extract (2); precipitated proteins (3); and Ab-free carrier (4). Note reciprocal precipitation in all three cases. **(D)** Extracts of B16 control (1, 4, 6) and profilin knock-out KO27 (2, 5, 7) cells were precipitated with immobilized Abs to profilin, followed by Western blotting and probing with Abs against profilin (Pfn), γ-tubulin (γ-Tb), GCP2, GCP4, actin, and calmodulin (negative control) as indicated. The sample lanes denoted 1–7 show: total cell extract, gel load (1, 2); no cell extract (3); precipitated proteins (4–5); and Ab-free carrier (6–7). Note co-precipitation of γ-tubulin, GCP2, GCP4, and actin with profilin Ab occurs in B16 cells solely. **(B, D)** Asterisks (*) in panels (B) and (D), lanes 2 and 3, and 3–5, respectively, denote cut off bands reflecting precipitating Abs.

centrosomes in KO27 cells (Fig 3C). To exclude a possible risk that the enhanced microtubule nucleation in the KO27 cells reflected an increase in nocodazole-resistant microtubules, B16 wild-type and KO27 cells were fixed in the presence of nocodazole and further processed for fluorescence microscopy using Abs to α-tubulin and γ-tubulin. The subsequent microscopy analysis revealed no difference in amount of nocodazole-resistant microtubules in B16 and KO27 cells (Fig S3A). We, therefore, conclude that microtubule nucleation from the centrosome is increased in absence of profilin. Consistently and in agreement with our earlier study (30), non–drug-treated KO27 cells displayed vastly more microtubules than wild-type cells (Fig S3B and C).

Because these observations pointed to the possibility that profilin in addition to its previously shown effect on microtubule elongation also may modulate de novo nucleation of microtubules from the centrosome, we decided to compare the nucleation rate of centrosomal microtubules in control and KO27 cells. To that end time-lapse imaging of cells expressing mNeonGreen-tagged microtubule end-binding protein 3 (EB3), decorating plus ends of the growing microtubules (37), was performed and followed by determination of the number of EB3 comets leaving the centrosomes per unit time, that is, the nucleation rate. Data analysis demonstrated a significantly augmented nucleation rate (2.2 times) in profilin-lacking cells (Fig 3D), which was consistent with the observation above of more microtubules in KO27 cells than in the control. The difference was particularly notable by comparing 10-frame projections of B16 control and KO27 cells expressing EB3 (Fig 3E).

Hence, it appeared that profilin is not only interfering with microtubule elongation as previously determined but is also a regulator of microtubule formation de novo. To provide further evidence for this conclusion, we performed phenotypic rescue experiments by co-expressing citrine-profilin or citrine-cathepsin B

(control) with EB1-tdTomato in KO27 cells, and citrine-cathepsin B with EB1-tdTomato in B16 wild-type cells. The expression of tagged proteins was documented by immunoblotting of whole-cell lysates (Fig 4A). As mentioned above, comparing the nucleation rate of microtubules from the centrosome in control and KO27 cells expressing citrine-cathepsin B revealed a significant increase (2.2 times) in the profilin-depleted cells, whereas the presence of citrine-profilin restored the nucleation rate in the KO27 cells to the level observed for control cells (Fig 4B). Collectively these data support the results obtained by measurement of the α-tubulin signal after nocodazole washout and recovery (Fig 3B) and provide strong arguments for profilin as a regulator of centrosomal microtubule nucleation. Furthermore, this puts profilin in a unique position as a coordinator of actin and microtubule organization in mammalian cells.

## Discussion

In this study we have continued to explore the role of the actin regulatory protein profilin for microtubule organization in mammalian cells. Previously, we demonstrated that a brief exposure to detergent before fixation enables immunofluorescence visualization of microtubule-associated profilin without disturbing fluorescence from the large pool of more generally distributed profilin in the cell (30). Here we applied this protocol to study the perinuclear distribution of profilin in mouse melanoma B16 cells and were able to detect profilin by immunofluorescence microscopy in centrosomes identified by γ-tubulin. We then took advantage of the fluorescent citrine-profilin (31) to independently establish that profilin distributes to centrosomes. Furthermore, reciprocal immunoprecipitation experiments combined with Western blotting demonstrated that profilin is a binding partner to components of

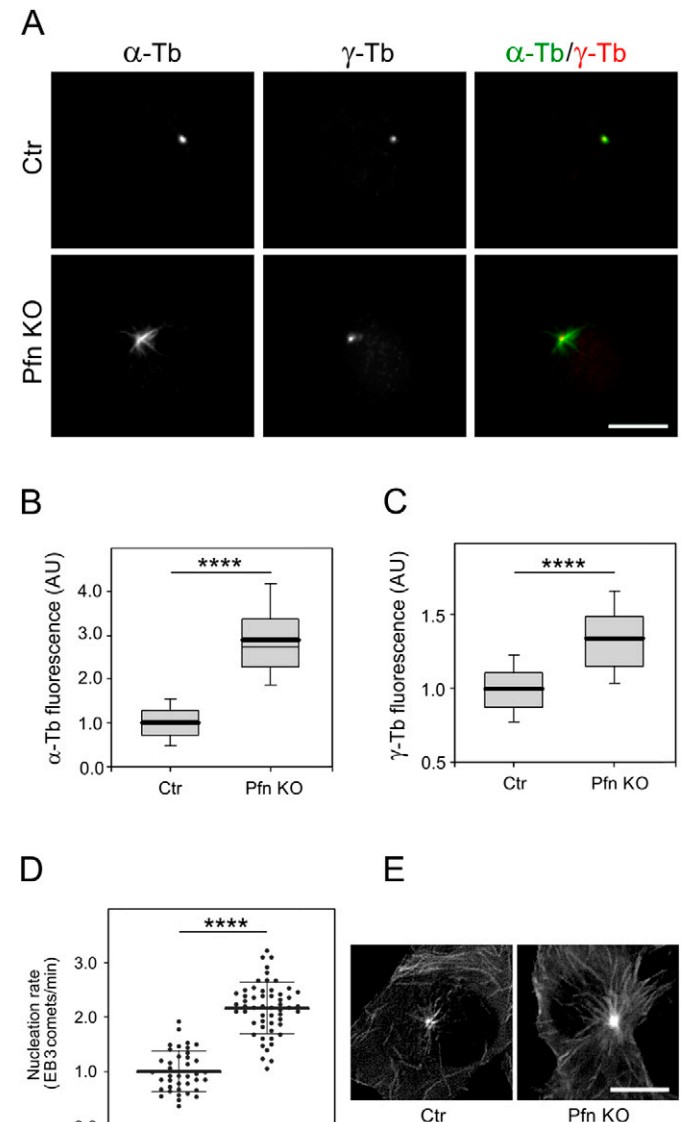

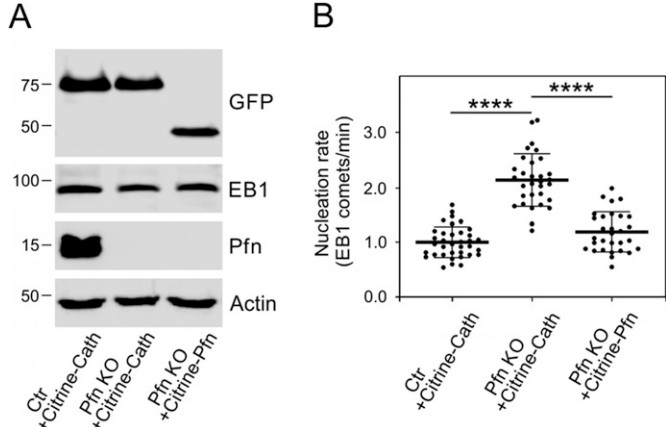

**Figure 4. Phenotypic rescue of increased microtubule nucleation in KO27 cells.** **(A)** Immunoblot analysis of whole-cell lysates of B16 control cells expressing citrine-cathepsin B (Ctr+Citrine-Cath), profilin knock-out KO27 cells expressing citrine-cathepsin B (Pfn KO+Citrine-Cath), and KO27 cells rescued by citrine-profilin (Pfn KO+Citrine-Pfn). Blots were probed with Abs to GFP to detect citrine-tagged proteins, to EB1 to detect co-expressed EB1-tdTomato, profilin (Pfn), and to actin (loading control). **(B)** Comparison of the microtubule nucleation rate observed in KO27 (Pfn KO) cells expressing Citrine-cathepsin B and Citrine-profilin, respectively, and in control cells (Ctr) expressing Citrine-cathepsin B. Three independent experiments and at least 10 cells analyzed in each experiment, n = 36 (Ctr+Citrine-Cath), n = 31 (Pfn KO+Citrine-Cath), and n = 30 (Pfn KO+Citrine-profilin); abbreviations as in left panel. The thick and thin lines within dot box plots represent mean ± SD; ****$P$ < 0.0001.

**Figure 3. Profilin modulates microtubule regrowth.** **(A)** B16 control and profilin knock-out (KO27) cells visualized after nocodazole washout and 2 min of recovery in drug-free medium at 37°C followed by processing for fluorescence labeling of microtubules (α-tubulin; α-Tb) and centrosomes (γ-tubulin; γ-Tb). Note the more prominent array of nascent microtubules extending from the centrosome in the KO27 cells compared with the control. Scale bar: 10 μm. **(B)** Box plots illustrating the distribution of α-tubulin fluorescence intensity (arbitrary units, AU) determined 3 min after drug washout at 28°C within a 1.0-μm region of interest in B16 control and KO27 cells, respectively; n = 330 (control; Ctr) and 322 (KO27; Pfn KO). **(C)** As in (B), but γ-tubulin fluorescence intensity; n = 339 (control) and 284 (KO27). The data in (B, C) are based on three independent experiments and >90 cells for each experimental condition. The thick and thin lines within each box represent the mean and median (50th percentile), respectively, whereas bottom and top represent the 25th and 75th percentiles. The whiskers below and above the box indicate the 10th and 90th percentiles; ****$P$ < 0.0001. **(D)** Microtubule nucleation rate in KO27 cells relative to control cells. Three independent experiments with at least 13 cells counted in each experiment, n = 39 (control) and 55 (KO27). The thick and thin lines within the dot box plots represent mean ± SD; ****$P$ < 0.0001. **(E)** Time-lapse imaging of control and KO27 cells expressing EB3-mNeonGreen. Tracks of EB3 comets captured during 10 s are shown. Scale bar: 10 μm.

the γ-tubulin ring complex. Finally, microtubule regrowth experiments and evaluation of de novo nucleation in wild-type B16 and the profilin-depleted B16 cells KO27 (31) disclosed that profilin is a negative regulator of microtubule nucleation.

We and others have shown that profilin interferes with microtubule elongation (30, 38). In our study by Nejedla et al (30), we presented data suggesting that the distribution of profilin along microtubules was dependent on formins. Because formins are well-known binding partners to profilin and have been demonstrated to attach to microtubules, we concluded that the association of profilin to microtubules was at least partially indirect via formins. Indeed, in support of an indirect interaction, in vitro experiments involving co-sedimentation and assembly assays did not reveal any interaction between purified profilin and polymerization-competent porcine brain tubulin in monomer or polymer form (30). However, in striking contrast, the subsequent study by Henty-Ridilla et al (38) reported that such a direct interaction in fact can occur. The reason for this discrepancy is unclear. Possibly it could be explained by variations of protein preparations or the different approaches used for analysis.

Combined with the current study, we have thus uncovered a dual role of profilin for microtubule organization in mammalian cells because it interferes with both centrosomal microtubule nucleation and microtubule elongation. Amazingly enough, profilin, therefore, operates in the molecular processes behind formation and growth of both actin filaments and microtubules; in the former case, it exert a positive role by providing actin as profilin-actin for polymer growth, whereas in the latter its function is to interfere with nucleation and elongation. Today, several proteins have been characterized as contributors to the connection and coordination of actin and microtubule organization (12, 39), but to our knowledge, the duality of profilin in this

respect is unique and puts profilin in a central position for the coupling of the actin and microtubule systems in mammalian cells.

Tentatively, the role of profilin in the centrosome may be related to recent observations that also actin is a component of this organelle. Profilin-actin serves as source of actin for microfilament polymerization and as such co-operates with actin nucleation and elongation promoting proteins (26, 27, 28, 29, 33, 39, 40). A possible scenario is therefore that the loss of profilin results in less of actin polymerization in the centrosomal region in analogy with what was recently reported for the cell edge (41); as a consequence, less of steric hindrance (8, 42) would reduce a possible space constrain for enhanced de novo microtubule nucleation. Alternatively, because actin is a structural component of functional γTuRCs (22, 23, 24), centrosomal profilin may tune the availability of actin for γTuRC. Therefore, profilin deletion could increase formation of functional γTuRC because of more actin being accessible for association with γTuRC. This in turn would enable enhanced de novo microtubule nucleation from centrosomes.

With this study, we emphasize profilin as a critical mediator of actin and microtubule cross-talk and extend the view of its role as a crucial component for cell architecture and behavior by its coordinated maintenance of actin and microtubule homeostasis as was recently discussed (39). We expect future studies to be directed to understand whether the interaction of profilin with either actin or components carrying the typical profilin-binding poly(L-proline)-motif (26, 27, 28, 29), or both is required for its inhibitory effect on centrosomal microtubule nucleation.

# Materials and Methods

### Antibodies

Mouse mAb TU-31 (IgG2b; for precipitation hybridoma supernatant diluted 1:2) to γ-tubulin, GCP2-01 (IgG2b; for precipitation hybridoma supernatant diluted 1:2) and GCP2-02 (IgG1; for immunoblotting hybridoma supernatant diluted 1:5) to GCP2 were described previously (43, 44, 45). Rabbit Abs to actin (# A2066; 1:10,000) and profilin I N-terminal (# P7749; 1:3,000 for immunoblotting, and 1:100 for fluorescence microscopy) (31), and mAb GTU-88 (IgG1; # T6557; 1:10,000 for immunoblotting, 1:1,000 for immunofluorescence) to γ-tubulin were from Sigma-Aldrich. Mouse mAb to GCP4 (IgG1, # sc-271876; 1:1,000) was purchased from Santa Cruz Biotechnology. Rabbit Abs to profilin (#ab50667; 1:200 for precipitation) and calmodulin (# ab45689; 1:5,000) were from Abcam. Rabbit Ab to α-tubulin (# 600-401-880; 1:100) was from Rockland. Rabbit Ab to non-muscle myosin heavy chain (# BT-561; 1:1,000; Biomedical Technologies) and mAbs NF-09 (IgG2a; hybridoma supernatant diluted 1:2) to neurofilament NF-M protein (46) served as negative controls in immunoprecipitation experiments. Antimouse and antirabbit Abs conjugated with HRP were from Promega Biotec (1:10,000). Antimouse Ab conjugated with DyLight 549 (1:1,000) or DyLight 649 (1:500) and antirabbit Ab conjugated with Alexa Fluor 488 were from Jackson Immunoresearch Laboratories (1:200).

### Cell cultures and transfection

Mouse melanoma B16-F1, control clone, and profilin KO clone 27 (B16 Pfn1, KO27) generated by Crispr/Cas9 (31) as well as human epithelial colorectal adenocarcinoma Caco-2 (HTB-37; ATCC) were cultured in DMEM (Thermo Fisher Scientific) supplemented with 10% FCS and antibiotics at 37°C in the presence of 5% $CO_2$. To prepare B16 cells expressing citrine-profilin 1 (31), citrine-cathepsin B (#56554; Addgene), or EB3-mNeonGreen (Allele Biotechnology), the cells were transfected with 2.5 μg plasmid DNA per 3-cm tissue culture dish using Lipofectamine LTX (Invitrogen) according to the manufacturer's instructions. The transfection mixture was replaced with fresh complete medium after 12 h followed by continued culturing for another 48 h after which fresh medium containing 1.2 mg/ml geneticin (G418; Sigma-Aldrich) was added, and the incubation was continued for 7 d.

For the phenotypic rescue experiment, B16 control or KO27 cells were co-transfected with two plasmids (total 2.5 μg DNA per 3-cm tissue culture dish, plasmid molar ratio 1:1) using Lipofectamine 3000 (Invitrogen) according to the manufacturer's instructions. After 24 h, the cells were seeded onto dishes for live cell imaging. B16 control cells were co-transfected with EB1-tdTomato (# 50825; Addgene) and citrine-cathepsin B (#56554; Addgene) plasmids. KO27 cells were co-transfected with EB1-tdTomato and citrine-cathepsin B plasmids or with EB1-tdTomato and citrine-profilin 1 plasmids.

### Immunoprecipitation

For immunoprecipitation experiments, washed cells were incubated for 10 min at 4°C with RIPA buffer (50 mM Tris, pH 8.0, 150 mM NaCl, 1% NP-40, 0.5% sodium deoxycholate, and 0.1% SDS) supplemented with protease inhibitors (Complete EDTA-free; Roche) and phosphatase inhibitors (1 mM $Na_3VO_4$, 1 mM NaF). The suspension was then centrifuged (20,000$g$, 10 min, 4°C) and the supernatant subjected to immunoprecipitation as described previously (47, 48) by incubation with Protein A beads (Protein A Sepharose CL-4B; GE-Healthcare Life Sciences) saturated with Abs as indicated (Figs 2 and S2). Gel electrophoresis and immunoblotting were performed using standard protocols (49). The HRP signal was detected with SuperSignal WestPico Chemiluminescent reagent (Pierce) and the LAS 3000 imaging system (Fujifilm).

### Microtubule regrowth

Microtubule regrowth from centrosomes was followed by nocodazole-washout experiments. Cells, cultured on coverslips were treated with nocodazole (Sigma-Aldrich) at a final concentration of 10 μM for 1 h at 37°C to depolymerize microtubules, then washed with PBS precooled to 4°C (three times, 5 min each) and, unless stated differently followed by regrowth for 3 min at 28°C in complete medium after which the cells were fixed and stained for α-tubulin and γ-tubulin as described previously (50). Briefly, cells cultured on coverslips were fixed in 3% formaldehyde, extracted with 0.5% Triton X-100 and post-fixed with cold methanol (F/Tx/M). The samples were then incubated sequentially with mAb to γ-tubulin (GTU-88) and rabbit Ab to α-tubulin followed by simultaneous labeling with secondary Abs being conjugated

with distinct fluorophores (DyLight 549-antimouse and AF488-antirabbit, respectively). The coverslips were mounted in MOWIOL 4-88 (Calbiochem) supplemented with 4,6-diamidino-2-phenylindole (DAPI; Sigma-Aldrich) and examined with a Delta Vision Core System (Applied Precission) equipped with a 60×/1.42 NA oil objective. Finally, the microtubule regrowth was determined by capturing the fluorescent signal in both channels from different areas per sample and the sum of fluorescence intensities, representing γ-tubulin and α-tubulin, respectively, was obtained from nine consecutive frames (0.2 µm steps), with the middle frame chosen with respect to the highest γ-tubulin intensity. Intensity quantification of a region of interest, defined as concentric circles of a radius of 1 µm and centered around the γ-tubulin marked centrosome was then done automatically using an in-house written macro for Fiji (51, 52).

### Fluorescence microscopy

Wide-field microscopy was performed using an Axiovert 200 M microscope (Carl Zeiss) equipped with a climate chamber, an EC-Plan-Neofluar 63×/1.4 objective lens, and a DG-4 light source (Sutter Instruments). Images were captured with a Cascade 1K camera (Roper). High-resolution confocal imaging was performed using an LSM800 AiryScan instrument (Carl Zeiss).

To simultaneously visualize γ-tubulin and citrine-profilin in B16 cells, permeabilization was performed with 10 µM digitonin (Calbiochem) in 25 mM Hepes buffer, pH 7.4, containing 2 mM EGTA, 115 mM CH₃COOK, 2.5 mM MgCl₂, and 150 mM sucrose (53) for 30 s, fixed with 3% formaldehyde in microtubule stabilizing buffer (50) for 20 min at room temperature and post-fixed with cold methanol at −20°C for 5 min. For γ-tubulin staining mAbs (GTU-88) and secondary Abs conjugated to DyLight 649 were used. Samples were mounted in Fluoromount-G (SouthernBiotech) and imaged using Andor Dragonfly 503 spinning disc confocal system (Oxford Instruments) with 40-µm pinhole size, equipped with Ixon Ultra 888 EMCCD 16 bit camera and 63×/1.4 NA oil objective. Consecutive z-stack images were captured with a step size of 0.1 µm and deconvoluted using Huygens Professional software v. 19.04 (Scientific Volume Imaging) with spinning disc module, SNR 20, and maximum iterations of 50, CMLE mode, and quality threshold set to 0.01. Citrine-tagged cathepsin B (56554; Addgene), which targets lysosomes served as negative control for citrine-profilin localization. The staining of γ-tubulin in cells expressing citrine-tagged cathepsin B revealed no association with centrosomes of the latter.

### Imaging of microtubule nucleation

B16 cells expressing EB3-mNeonGreen were cultured in a 35 mm µ-Dish with ibidi polymer coverslip bottom (Ibidi GmbH); 30 min before imaging, the medium was replaced with FluoroBrite DMEM (Thermo Fisher Scientific), supplemented with 25 mM Hepes and 1% FCS. Time-lapse sequences were collected in three optical slices (0.13 µm steps) for 1 min at 1 s interval with the Andor Dragonfly 503 spinning disc confocal system (Oxford Instruments) equipped with a stage top microscopy incubator (Okolab), a 488 nm solid-state 150 mW laser, HCX PL APO 63× oil objective, NA 1.4, and a Zyla sCMOS 16 bit camera. For each experiment, at least 10 cells were imaged using the following acquisition parameters: 40-µm

pinhole size, 15% laser power, 150-ms exposure time, and an 525/50-nm emission filter. The time-lapse sequences were deconvoluted with Huygens Professional software version 19.04 (Scientific Volume Imaging), and maximum intensity projection of z stacks was made for each time point in Fiji. Newly nucleated microtubules were detected by manual counting of EB3 comets emanating from the centrosomes. For time-lapse imaging in phenotypic rescue experiments, B16 cells co-expressing EB1-tdTomato with citrine-profilin 1 or citrine-cathepsin B were cultivated and imaged as described above using a 561 nm solid-state 100 mW laser and an Ixon Ultra 888 EMCCD 16 bit camera. For each experiment, at least 10 cells, expressing both EB1-tdTomato and citrine-tagged protein at a comparable level, were imaged using the following acquisition parameters: 40-µm pinhole size, 15% laser power, 150-ms exposure time, and an 600/50-nm emission filter. Reference still images of citrine-tagged proteins were obtained with a 488 nm solid-state 150 mV laser using the acquisition parameters: 40-µm pinhole size, 15% laser power, 150-ms exposure time, and an 525/50-nm emission filter. Deconvolution was performed as described above.

### Statistics

Significance was tested using a two-tailed, unpaired *t* test or one-way ANOVA followed by a Sidak's post hoc test using Prism 8 software (GraphPad Software). For all analyses, *P*-values were represented as follows: **$P < 0.01$; ***$P < 0.001$; ****$P < 0.0001$.

# Supplementary Information

# Acknowledgements

We acknowledge support to P Dráber from the Grant Agency of the Czech Republic, grant 18-27197S and by institutional research support (RVO 68378050); to A Klebanovych from the Charles University, grant GA UK 142618; to R Karlsson from Carl Tryggers Stiftelse (CTS 17:248) and to M Nejedlá by long-term development project (RVO 67985939). Microscopy studies were performed at the imaging facility at Stockholm University (IFSU) and at the Microscopy Center of the Institute of Molecular Genetics (grant LM2018129 from MEYS). We thank Dr. Klemens Rottner, Braunschweig, for critical reading of the manuscript before submission.

### Author Contributions

M Nejedlá: formal analysis, investigation, and writing—review and editing.
A Klebanovych: formal analysis and investigation.
V Sulimenko: investigation.
T Sulimenko: formal analysis and investigation.
E Dráberová: investigation.
P Dráber: conceptualization, resources, formal analysis, supervision, validation, project administration, and writing—review and editing.

R Karlsson: conceptualization, resources, supervision, investigation, project administration, and writing—original draft, review, and editing.

## Conflict of Interest Statement

The authors declare that they have no conflict of interest.

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
