## [Reviewer comments · Life Science Alliance]

Life Science Alliance

The Actin regulator Profilin 1 is Functionally Associated with the Mammalian Centrosome

Michaela Nejedlá, Anastasiya Klebanovych, Vadym Sulimenko, Tetyana Sulimenko, Eduarda Dráberová, Pavel Dráber, and Roger Karlsson

DOI: <https://doi.org/10.26508/lsa.202000655>

Corresponding author(s): Roger Karlsson, Stockholm University and Pavel Dráber, Laboratory of Biology of Cytoskeleton

Review Timeline:

Submission Date:	2020-01-22
Editorial Decision:	2020-02-17
Revision Received:	2020-10-11
Editorial Decision:	2020-10-23
Revision Received:	2020-10-29
Accepted:	2020-11-02

Scientific Editor: Shachi Bhatt

Transaction Report:

February 17, 2020

Re: Life Science Alliance manuscript #LSA-2020-00655-T

Prof. Roger Karlsson
Stockholm University
Molecular Biosciences, The Wenner-Gren Institute
Svante Arrhenius väg 20C
Stockholm SE 10691
Sweden

Dear Dr. Karlsson,

Thank you for submitting your manuscript entitled "The Actin regulator Profilin 1 is Functionally Associated with the Mammalian Centrosome". The manuscript has been evaluated by expert reviewers, whose reports are appended below.

As you will see, the reviewers think a specific function of profilin 1 at centrosomes is not supported by the data provided. They further note missing controls, quantifications and rescue experiments, and that the suggested effect on MT stability is not supported by the data shown.

We discussed your work in light of the reviewer input. Although your manuscript is intriguing, we feel that the points raised by the reviewers, and especially the one pertaining to lack of support for a functional role of profilin 1 at centrosomes, are more substantial than can be addressed in a typical revision period. If you wish to expedite publication of the current data, it may be best to pursue publication at another journal and we are thus returning your manuscript to you.

Given the interest in the topic, we would however be open to resubmission to Life Science Alliance of a significantly revised and extended manuscript that fully addresses the reviewers' concerns and provides better support for a functional role at centrosomes. Such a revised version would be subject to further peer-review.

Regardless of how you choose to proceed, we hope that the comments below will prove constructive as your work progresses. We would be happy to discuss the reviewer comments further once you've had a chance to consider the points raised in this letter.

Thank you for thinking of Life Science Alliance as an appropriate place to publish your work.

Sincerely,

Andrea Leibfried, PhD
Executive Editor
Life Science Alliance
Meyerhofstr. 1
69117 Heidelberg, Germany
t +49 6221 8891 502
e a.leibfried@life-science-alliance.org

Reviewer #1 (Comments to the Authors (Required)):

In this short paper from Nejedla et al., the authors postulate a role for profilin at the centrosome to regulate microtubule (MT) dynamics and subsequent cell cycle progression. The findings are of interest in particular in light of the multiple recent evidences for a function for actin at the centrosome. Some of the data presented appear rather preliminary. I have some suggestions on how to improve the manuscript in order to strengthen the authors conclusions.

The authors claim that the absence of profilin 1 leads to excessive MT growth, based on a-tubulin staining in 3A. They write that profilin 1 therefore controls MT stability. The authors could test this by measuring GFP-EB3 dynamics in the profilin KO cells in order to substantiate this claim. This should be a straight forward and robust experiment.

The authors should try to confirm their findings from the profilin KO cells using siRNA to exclude a clonal selection phenotype. This would also test the relevance in other cell lines perhaps.

Figure 3D- are the KO cells bigger? The described observation of the "vastly more densely packed microtubule arrays " needs quantification.

I find Figure 4 is very weak. One cannot discriminate the "multinucleated" phenotype (the legend title has a spelling error as well). In fact this is not even mentioned in the results section nor is figure 4 well explained there. I don't understand the zoom in 4A. There seems to be very little or no additional information in these insets.

Rescue experiments in the profilin KO cells should be performed (in principle for all observations). The authors conclude that profilin KO cells grow slower due to the absence of profilin 1, in which case reintroducing profilin 1 should revert this phenotype. This should be tested. They could also measure cell growth in a more unbiased way rather than counting a few cells (e.g. FACS based assays for cell cycle stages). Can they exclude a apoptosis phenotype?

Reviewer #2 (Comments to the Authors (Required)):

The work of Nejedla and colleagues about profilin localisation and activity at the centrosome is novel and interesting. It is timely since several papers have discussed recently the role of actin filaments at the centrosome but the regulation of actin filament growth there is still poorly described. However the manuscript requires few additional controls to fully support the conclusions that were formulated.

- « microtubule array reformed more rapidly in cells lacking profilin 1 »

The author also said that microtubules were more stable in these cells, suggesting that the pictures shown in figure 3A may show nocodazole-resistant microtubules rather than newly assembled microtubules after the drug washout. As a control, authors should make sure that microtubules were disassembled to the same extent in the two cell lines.

- « KO27 cells displayed an increased microtubule stability »

The extent of the network shown in figure 3D is not an evidence of microtubule stability. A nocodazole-resistance or cold-resistance assay should be performed.

- « our observations here [...] suggest that profilin-actin is a principal source of actin for centrosome-derived filament formation »

This is a strong and potentially significant statement. But the existence of such centrosome-derived filaments is not shown in the manuscript. Unfortunately, this concern also applies to many recent papers in this field, putting some doubts on the existence of such a network. This network should at least be shown in order to be discussed. It would be even better of course if the role of profilin on the size/density of the network could be directly measured.

Reviewer #3 (Comments to the Authors (Required)):

How the actin and microtubule cytoskeletons interact with each other is an important question for understanding cellular behavior and has been a very exciting recent avenue of research. This work follows up on the observation from this group and others that the actin regulator profilin can affect the microtubule cytoskeleton. Specifically the authors identify a novel localization of profilin to the centrosome, show that profilin and gamma tubulin can be coimmunoprecipitated from cell extracts. They then attempt to link their localization and interaction data with data showing that loss of profilin leads to increased microtubule regrowth from centrosomes, cell populations with an increase in the average number of centrosomes, and a decrease in cell numbers.

While the individual experiments appear to be performed well, I have some major concerns with the interpretation of the data. Particularly in how the experiments in figures 3 and 4 are interpreted. These experiments show phenotypes associated with the loss of profilin. However, the linkage of these phenotypes to the specific loss of profilin function at the centrosome is more suggestive and not definitive. The data presented is consistent with, but not demonstrative of, profilin being "functionally associated with the mammalian centrosome" as their title indicates. As discussed in more detail below, there are non-centrosomal explanations, from the work of this group and others, for all of the phenotypes they observe in the absence of profilin. If the authors want to assert that they have demonstrated specific functions of profilin at the centrosome, then significant experiments would need to be performed to pinpoint that the phenotypes they report are specifically a consequence of profilin function at the centrosome. Otherwise, significant rewriting of the results and conclusions to reflect what the experiments definitively show, remove conclusions that the data does not support and replace these conclusions with discussion of the breadth of possible models supported by the data would be advised. I provide some examples below.

Major concerns:

1) In figure 3 the authors perform a microtubule regrowth assay in the presence and absence of profilin, showing increase amount of microtubules around centrosomes after regrowth when profilin has been knocked out. They also see an elevated amount of gamma tubulin at centrosomes in the profilin knock out. From this experiment the authors conclude this caused by "affecting gamma TURC function" and that "profilin can also modulate the de novo nucleation of microtubules". While their data is consistent with these statements, their data does not demonstrate an effect of profilin on gamma TURC function at the centrosome or on microtubule nucleation. These results show an effect of profilin on gamma tubulin localization and an effect on the amount of centrosome-associated microtubule polymer after a short regrowth. As presented, this data is entirely consistent with the effect being the result of other non-centrosomal functions of profilin on

microtubules. Both this group and the Goode group have shown that in cells lacking profilin, microtubules grow at a faster rate. The increase in microtubule polymer measured in Figure 3 could be the result of the previously observed increase in the rate of polymerization seen in the absence of profilin. To demonstrate that profilin is effecting the nucleation of microtubules at centrosomes, the authors would need to perform additional experiments to demonstrate that more individual microtubules were being generated, not just more polymer. For example, the authors could count the number of EB1 comets that emerge per minute from centrosomes following drug/cold release. An increase in the number of EB1 comets per minute would be a firmer indication of an increase in microtubule nucleation in the absence of profilin.

2) In figure 4 the authors show that profilin knockdown cells have increased numbers of centrosomes, most notably an increase in the number of cells with 2 centrosomes and a small fraction with 3 or more. Furthermore they show a decrease in the number of cells after 72 hours of growth. This result is discussed in one sentence in the text and the relation of this result to the rest of the text is not clear. These results could arise for any number of reasons, which may or may not be related to profilin at the centrosome. One hypothesis is that in the absence of profilin, the cells are spending more time in G2 than in controls. This would result in an increased frequency in cells with 2 centrosomes. The cells lacking profilin appear to have larger nuclei than the controls, which would also be consistent with a longer G2. A delay in G2 could also explain the slow growth of these cells. While experiments to address the cell cycle in these cells would be useful to understand the change in the numbers of centrosomes that occurs in when profilin is absent, it would take additional experiments to link this phenotype specifically to the function of profilin at the centrosome rather than an affect of profilin loss elsewhere in the cell.

3) The title for the legend for figure 4 is "Cells lacking profilin 1 show prolonged cell cycle and becomes multinucleated". The data in the figure do not show either of these results. All of the cells appear mononucleated. As discussed above, this data is consistent with a prolonged cell cycle, but it does not demonstrate a prolonged cell cycle.

4) The final sentence of the paper -

I do not believe the data in this paper provides sufficient evidence for most of this sentence.

A) "Our observations here of profilin negatively regulating centrosomal microtubule nucleation/elongation" - See major concern 1.

B) "impacting on cell progression through G2 and mitosis" - See major concern 3.

C) "suggest that profilin-actin is a principal source of actin for centrosome-derived filament formation" - It is possible that this is the case, but this paper does not provide any evidence to support this statement. As the authors mention, Arp2/3 is thought to be the major source of actin at the centrosome. However, the Bear group has shown that in the absence of Profilin 1 there was an increase in the amount of Arp2/3 and actin in lamellipodia. Overexpression of profilin had the opposite effect. This might suggest that rather than be the principal source for actin at the centrosome, profilin actin might be inhibitory towards centrosomal actin formation.

D) "making profilin a crucial component of actin and microtubule cross-talk also at the centrosome." - While the authors show profilin is at the centrosome and can interact with gamma tubulin, the rest of their data does not point specifically to profilin function at the centrosome as opposed to elsewhere in the cell.

Minor concerns:

1) In Figure 1 there appears to be some variation in the amount of profilin on centrosomes. For example in 1A profilin appears to be stronger on one centrosome and possibly absent on the other. Is profilin seen at all centrosomes? The authors should report the number of cells observed and the number of centrosomes where they observed profilin.

- 2) In the legend of Figure 1 the authors say cells were captured at "different cell cycle stages". While I infer that they are showing G1, G2 and metaphase, the authors should indicate in the legend or figure which stages the cells are at.
- 3) In the reciprocal IPs in Figure 2 the authors should clarify what "gel load" refers to. While I infer that this is a blot of the starting material for the IP experiment, what this is should be specified.
- 4) In the discussion of the results for Figure 2 the authors state "The same was also observed for GCP4 (not shown), all together being consistent with the possibility that not only profilin but also the profilin-actin complex is located to the centrosome,...". I am unclear why this experiment supports the conclusion the profilin-actin is involved more than the previous experiment did. Either clarify why poly-proline pull down is more consistent with profilin-actin complex than profilin pull down or remove this statement.
- 5) In figure 2 the authors show poly-proline binding results in precipitation of both profilin and gamma-tubulin complex. Do the authors have evidence that gamma-TURC alone does not bind poly proline and that it depends on profilin? Perhaps from an IP from their profilin KO cells.
- 6) The text referring to the microtubule regrowth assay in figure 3 indicates that the cells were observed after 2 minutes of recovery, while the legend and methods indicate 3 minutes.

Once more we express our thankfulness to the reviewers for an excellent review of our Manuscript. Our answers to their criticism and suggestions are detailed point-by-point below along with a specification of the changes introduced in the new version (highlighted in yellow). A summary of the major alterations are as follows (also included in the "letter to the editor").

- Previous Fig 2 now includes a new Panel D, illustrating IP-results from knock-out cells as requested (referee #3);
- Previous Fig 3, panel D is transferred to Fig S3B. Instead two new panels (D-E) are included to Fig 3 to illustrate the result of experiments where the centrosomal MT-nucleation was determined (referees # 1 & 3)
- Previous Fig 4 and the corresponding text has been removed. Instead Fig 4 in the new manuscript illustrates the rescue experiment of centrosomal MT-nucleation mentioned above (referee #1).
- Previous Fig S1 is now shifted to Fig S2. In the current version the new Fig S1 illustrates the presence of Pfn in centrosomes during interphase and metaphase (referee #3).
- Current version of Fig S2 is identical to previous Fig S1.
- Current version of Fig S3A displays nocodazole-resistant MTs in control and profilin knock-out cells after nocodazole treatment (referee #2). Fig S3B is identical to Fig 3D in the previous version and along with the newly added Fig S3C illustrates the increased density of MTs in Pfn 1-lacking cells.

Unfortunately the pandemic, which rapidly progressed soon after we received the referee report led to lengthy delays in the refinement work but now at the stage of resubmission we conclude that our manuscript has been significantly improved.

Thank you

Reviewer #1 (Comments to the Authors (Required)):

In this short paper from Nejedla et al., the authors postulate a role for profilin at the centrosome to regulate microtubule (MT) dynamics and subsequent cell cycle progression. The findings are of interest in particular in light of the multiple recent evidences for a function for actin at the centrosome. Some of the data presented appear rather preliminary. I have some suggestions on how to improve the manuscript in order to strengthen the authors conclusions.

We appreciate the positive attitude and suggestions expressed by the referee.

The authors claim that the absence of profilin 1 leads to excessive MT growth, based on α -tubulin staining in 3A. They write that profilin 1 therefore controls MT stability. The authors could test this by measuring GFP-EB3 dynamics in the profilin KO cells in order to substantiate this claim. This should be a straight forward and robust experiment.

This is an excellent suggestion and we used the approach to document and quantitate "EB3-comets" emerging from centrosomes in control and Pfn 1 knock out cells (KO27) by directly evaluating microtubule nucleation. The results from this analysis are presented in new Figs 3D & E. Furthermore we realized that our original formulation "MT stability" was unfortunate and misleading; therefore it has been removed and the text altered in accordance.

The authors should try to confirm their findings from the profilin KO cells using siRNA to exclude a clonal selection phenotype. This would also test the relevance in other cell lines perhaps.

We find this superfluous since our previous observation (see Ref # 30) has been independently repeated by another group using a non-related cell line (Henty-Ridilla et al *Curr Biol* 27: 3535-3543; Ref #34 in the new manuscript), and since the IP-experiment demonstrating the profilin- γ TuRC interaction (Fig 2) was repeated with Caco-cells (Figs S2A-B in the current manuscript), suggesting that the phenomenon is general.

Figure 3D- are the KO cells bigger? The described observation of the "vastly more densely packed microtubule arrays" needs quantification.

Concerning the cell size, the KO-cells show what appears to be a larger variation in size, which may relate to some of them becoming multinucleate. However, we cannot exclude that the KO-cells attach to the substratum differently and therefore becomes "flatter" but not necessarily of bigger size (i.e. larger volume)

Concerning the density of the MT-array, we present a striking difference in α -Tubulin fluorescence in the KO-cells (in current manuscript, Figs S3B-C). This is described on page 6 (bottom) and page 7 (top). The wordings pointed out by the referee have been altered in the new version.

I find Figure 4 is very weak. One cannot discriminate the "multinucleated" phenotype (the legend title has a spelling error as well). In fact this is not even mentioned in the results section nor is figure 4 well explained there. I don't understand the zoom in 4A. There seems to be very little or no additional information in these insets.

Figure 4 from our previous submission has been removed and the text altered in accordance.

Rescue experiments in the profilin KO cells should be performed (in principle for all observations). The authors conclude that profilin KO cells grow slower due to the absence of profilin 1, in which case reintroducing profilin 1 should revert this phenotype. Can they exclude a apoptosis phenotype?

Rescue experiments as suggested have been performed and the results, demonstrating reversal of the increased MT nucleation after reintroducing profilin by expression of the chimeric citrine-profilin construct in knock-out cells are described at page 7 and in the newly added Figs 4A & 4B. Also relevant to this comment by the referee is our previous study ref #31 (Nejedla et al 2017), which describes rescue by citrine-Pfn of several other aspects of the knock-out phenotype, i.e. F-actin distribution, migratory properties, increased tubulin acetylation and increased Pfn2 expression (see Fig 5 in ref #31).

In the currently submitted manuscript, we concentrated on MT nucleation and since the comment concerning cell growth and apoptosis is related to the previous Fig 4, which now is removed (see above) this comment is irrelevant to the new submission.

Reviewer #2 (Comments to the Authors (Required)):

The work of Nejedla and colleagues about profilin localisation and activity at the centrosome is novel and interesting. It is timely since several papers have discussed recently the role of actin filaments at the centrosome but the regulation of actin filament growth there is still poorly described. However the manuscript requires few additional controls to fully support the conclusions that were formulated.

We are pleased to learn that the referee finds our manuscript novel and timely

- « microtubule array reformed more rapidly in cells lacking profilin 1 »

The author also said that microtubules were more stable in these cells, suggesting that the pictures shown in figure 3A may show nocodazole-resistant microtubules rather than newly assembled microtubules after the drug washout. As a control, authors should make sure that microtubules were disassembled to the same extent in the two cell lines.

This is an important point; consequently the suggested experiment was performed showing no difference with respect to nocodazole-resistant MTs between control and Pfn knock-out cells (see text page 6, bottom and the newly added Fig S3A).

- « KO27 cells displayed an increased microtubule stability »

The extent of the network shown in figure 3D is not an evidence of microtubule stability. A nocodazole-resistance or cold-resistance assay should be performed.

The quoted expression referring to MT stability was misleading. It is removed in the current manuscript. The striking difference in density of the MT-array between the control and Pfn 1 knock-out cells as demonstrated by α -tubulin fluorescence is described in the current manuscript on and page 7 (top) and further illustrated in (Figs S3B-C).

- « our observations here [...] suggest that profilin-actin is a principal source of actin for centrosome-derived filament formation »

This is a strong and potentially significant statement. But the existence of such centrosome-derived filaments is not shown in the manuscript. Unfortunately, this concern also applies to many recent papers in this field, putting some doubts on the existence of such a network.

This network should at least be shown in order to be discussed. It would be even better of course if the role of profilin on the size/density of the network could be directly measured.

The original statement has been modified (last paragraphs page 7 & 8). However, we consider a discussion involving references to articles describing centrosome derived actin filaments as highly relevant (e.g. Refs 8 & 42 in the new manuscript version). This view is also reflected by referee #1's introductory comment "The findings are of interest in particular in light of the multiple recent evidences for a function for actin at the centrosome". In our opinion therefore, the modified discussion at the end of the current manuscript concerning possible mechanisms behind our observations is well motivated.

Reviewer #3 (Comments to the Authors (Required)):

How the actin and microtubule cytoskeletons interact with each other is an important question for understanding cellular behavior and has been a very exciting recent avenue of research. This work follows up on the observation from this group and others that the actin regulator profilin can affect the microtubule cytoskeleton. Specifically the authors identify a novel localization of profilin to the centrosome, show that profilin and gamma tubulin can be coimmunoprecipitated from cell extracts. They then attempt to link their localization and interaction data with data showing that loss of profilin leads to increased microtubule regrowth from centrosomes, cell populations with an increase in the average number of centrosomes, and a decrease in cell numbers.

While the individual experiments appear to be performed well, I have some major concerns with the interpretation of the data. Particularly in how the experiments in figures 3 and 4 are interpreted. These experiments show phenotypes associated with the loss of profilin. However, the linkage of these phenotypes to the specific loss of profilin function at the centrosome is more suggestive and not definitive. The data presented is consistent with, but not demonstrative of, profilin being "functionally associated with the mammalian centrosome" as their title indicates. As discussed in more detail below, there are non-centrosomal explanations, from the work of this group and others, for all of the phenotypes they observe in the absence of profilin. If the authors want to assert that they have demonstrated specific functions of profilin at the centrosome, then significant experiments would need to be performed to pinpoint that the phenotypes they report are specifically a consequence of profilin function at the centrosome. Otherwise, significant rewriting of the results and conclusions to reflect what the experiments definitively show, remove conclusions that the data does not support and replace these conclusions with discussion of the breadth of possible models supported by the data would be advised. I provide some examples below.

We are most thankful for the valuable criticism and suggestions made by this referee and have performed new experiments, which have resulted in a strong case for profilin having a functional role at the centrosome. Particularly the phenotypic rescue experiment where citrine-profilin restored centrosomal MT-nucleation to control level in knock-out cells is a robust argument for this.

Major concerns:

1) In figure 3 the authors perform a microtubule regrowth assay in the presence and absence of profilin, showing increase amount of microtubules around centrosomes after regrowth when profilin has been knocked out. They also see an elevated amount of gamma tubulin at centrosomes in the profilin knock out. From this experiment the authors conclude this caused by "affecting gamma TURC function" and that "profilin can also modulate the de novo nucleation of microtubules". While their data is consistent with these statements, their data does not demonstrate an effect of profilin on gamma TURC function at the centrosome or on microtubule nucleation.

These results show an effect of profilin on gamma tubulin localization and an effect on the amount of centrosome-associated microtubule polymer after a short regrowth. As presented, this data is entirely consistent with the effect being the result of other non-centrosomal functions of profilin on microtubules. Both this group and the Goode group have shown that in cells lacking profilin, microtubules grow at a faster rate. The increase in microtubule polymer measured in Figure 3 could be the result of the previously observed increase in the rate of polymerization seen in the absence of profilin.

To demonstrate that profilin is effecting the nucleation of microtubules at centrosomes, the authors would need to perform additional experiments to demonstrate that more individual microtubules were being generated, not just more polymer. For example, the authors could count the number of EB1 comets that emerge per minute from centrosomes following drug/cold release.

An increase in the number of EB1 comets per minute would be a firmer indication of an increase in microtubule nucleation in the absence of profilin.

Point taken; in the new manuscript we demonstrate a variation in nucleation rate by using the approach with a fluorescent end-binding protein as suggested. The result is presented in the newly added Figs 3D-E along with measurements of α -tubulin fluorescence intensity after drug removal from control and Pfn knock-out cells (see text page 6 & 7). The data presented here and in the following section, dealing with the rescue experiment (new Fig 4 in current manuscript), demonstrate in our view a direct influence of centrosomal profilin on MT-nucleation and growth from the centrosome (see further below)

2) In figure 4 the authors show that profilin knockdown cells have increased numbers of centrosomes, most notably an increase in the number of cells with 2 centrosomes and a small fraction with 3 or more. Furthermore they show a decrease in the number of cells after 72 hours of growth. This result is discussed in one sentence in the text and the relation of this result to the rest of the text is not clear. These results could arise for any number of reasons, which may or may not be related to profilin at the centrosome. One hypothesis is that in the absence of profilin, the cells are spending more time in G2 than in controls.

This would result in an increased frequency in cells with 2 centrosomes. The cells lacking profilin appear to have larger nuclei than the controls, which would also be consistent with a longer G2. A delay in G2 could also explain the slow growth of these cells.

While experiments to address the cell cycle in these cells would be useful to understand the change in the numbers of centrosomes that occurs in when profilin is absent, it would take additional experiments to link this phenotype specifically to the function of profilin at the centrosome rather than an affect of profilin loss elsewhere in the cell.

We agree with the referee and consider this possible function of profilin to be subject for a separate study. Consequently, Fig 4 from the previous manuscript version and the related text is not included in the present current manuscript.

3) The title for the legend for figure 4 is "Cells lacking profilin 1 show prolonged cell cycle and becomes multinucleated". The data in the figure do not show either of these results. All of the cells appear mononucleated. As discussed above, this data is consistent with a prolonged cell cycle, but it does not demonstrate a prolonged cell cycle.

See above

4) The final sentence of the paper -

I do not believe the data in this paper provides sufficient evidence for most of this sentence.

A) "Our observations here of profilin negatively regulating centrosomal microtubule nucleation/elongation" - See major concern 1.

We refer to the new data included, which are dealt with in response to "major concern 1" above.

B) "impacting on cell progression through G2 and mitosis" - See major concern 2.

The new submission does not contain the criticized passage, see above "major concern 2"

C) "suggest that profilin-actin is a principal source of actin for centrosome-derived filament formation" - It is possible that this is the case, but this paper does not provide any evidence to support this statement. As the authors mention, Arp2/3 is thought to be the major source of actin (he/she must mean "major actin nucleation factor") at the centrosome. However, the Bear group has shown that in the absence of Profilin 1 there was an increase in the amount of Arp2/3 and

actin in lamellipodia. Overexpression of profilin had the opposite effect. This might suggest that rather than be the principal source for actin at the centrosome, profilin actin might be inhibitory towards centrosomal actin formation.

The original text discussing this subject has been modified substantially, see page 7 & 8 in current manuscript, and we hope our new version pointing to two possible mechanisms of profilin at the centrosome based on our observations is satisfactory. Nonetheless, we like to point out that 1) Arp2/3 to our knowledge cannot be a source of actin; 2) profilin will interfere with polymerization by actin sequestration if there are no nucleation factors or free barbed filament ends available, i.e. profilin is part of the regulatory mechanism; 3) the Mullins-lab has showed that also Arp2/3 can operate with profilin:actin as source of actin; 4) Arp2/3 requires WASP/WAVE-related nucleation promoting factors to operate, meaning that we are back to the starting point for the reasoning again: what is the source of actin – what keeps actin polymerization-competent (ATP-loaded) without starting to polymerize unless filament formation is called for; 5) the recent paper by Skruber et al 2020 (ref #41 in the current manuscript), reports requirement of profilin for Arp2/3 and Vasp-dependent actin polymerization. Together with the new data in our current version the manuscript, we therefore consider profilin as a strong candidate to operate with actin as profilin-actin in the context of our observations as we discuss towards the end of the manuscript.

D) "making profilin a crucial component of actin and microtubule cross-talk also at the centrosome." - While the authors show profilin is at the centrosome and can interact with gamma tubulin, the rest of their data does not point specifically to profilin function at the centrosome as opposed to elsewhere in the cell.

The current data presentation in Figs 3 and 4 strongly argues in favor for a direct function of profilin at the centrosome.

Minor concerns:

1) In Figure 1 there appears to be some variation in the amount of profilin on centrosomes. For example in 1A profilin appears to be stronger on one centrosome and possibly absent on the other. Is profilin seen at all centrosomes? The authors should report the number of cells observed and the number of centrosomes where they observed profilin.

The referee points to an intensity variation that is likely to be due to technical rather than biological reasons, e.g. it is possible that the two centrosomes were captured at different z-positions by the confocal microscopy used. Although, the fluorescence intensity of Pfn in the centrosomes varies (newly added Fig S1) we have never noted Pfn to be absent (not by immunofluorescence labeling nor by expression citrine-profilin) from this organelle. Although we cannot exclude that the two centrosomes at some stage during the G2 phase varies in this respect we have not noted such a variation.

2) In the legend of Figure 1 the authors say cells were captured at "different cell cycle stages". While I infer that they are showing G1, G2 and metaphase, the authors should indicate in the legend or figure which stages the cells are at.

We have followed the referees advice and indicated the cell cycle stages when appropriate

3) In the reciprocal IPs in Figure 2 the authors should clarify what "gel load" refers to. While I infer that this is a blot of the starting material for the IP experiment, what this is should be specified.

The referee is correct, "gel load" refers to starting material (total cell extract). This is specified in the Figure legend.

4) In the discussion of the results for Figure 2 the authors state "The same was also observed for GCP4 (not shown), all together being consistent with the possibility that not only profilin but also the profilin-actin complex is located to the centrosome,...". I am unclear why this experiment supports the conclusion the profilin-actin is involved more than the previous experiment did. Either clarify why poly-proline pull down is more consistent with profilin-actin complex than profilin pull down or remove this statement.

5) In figure 2 the authors show poly-proline binding results in precipitation of both profilin and gamma-tubulin complex. Do the authors have evidence that gamma-TURC alone does not bind poly proline and that it depends on profilin? Perhaps from an IP from their profilin KO cells.

The statement along with the description of the poly-proline pull down and the data presentation referring to experiments with the PLP-Sepharose has been removed

As suggested by the referee we performed new IP experiments from profilin KO cells, see pages 5 (bottom) and 6 (top), and the newly added Fig 2D.

6) The text referring to the microtubule regrowth assay in figure 3 indicates that the cells were observed after 2 minutes of recovery, while the legend and methods indicate 3 minutes.

This is correctly noted. However, in addition to the recovery time, the experiments were performed under slightly different temperature conditions (2 min at 37 C, and 3 min at 28 C) as indicated. The reason for this difference is that our two groups involved in the study used slightly different protocols. Therefore it is stated in Methods that "unless stated differently ... 3 min at 28 C" (page 10). Also in the legend to Fig 3 the conditions are specified "...2 min at 37 C" (panel A) and "3 min at 28 C" (panel B). We see no qualitative impact on the result due to the different conditions, but naturally the reader should be properly informed about this fact.

October 23, 2020

RE: Life Science Alliance Manuscript #LSA-2020-00655-TR-A

Prof. Roger Karlsson
Stockholm University
Molecular Biosciences, The Wenner-Gren Institute
Svante Arrhenius väg 20C
Stockholm SE 10691
Sweden

Dear Dr. Karlsson,

Thank you for submitting your revised manuscript entitled "The Actin regulator Profilin 1 is Functionally Associated with the Mammalian Centrosome". We would be happy to publish your paper in Life Science Alliance pending minor requests from reviewers and final revisions necessary to meet our formatting guidelines.

Along with the points listed below, please also attend to the following:

- please add ORCID ID for secondary corresponding author-you should have received instructions on how to do so
- please provide your manuscript text in editable doc format
- please separate the Results and Discussion section into separate Results and a separate Discussion section

A. FINAL FILES:

B. MANUSCRIPT ORGANIZATION AND FORMATTING:

Sincerely,

Shachi Bhatt, Ph.D.
Executive Editor
Life Science Alliance
<https://www.lsjournal.org/>
Tweet @SciBhatt @LSAJournal

Reviewer #2 (Comments to the Authors (Required)):

The authors addressed experimentally most of the comments I raised and honestly discussed the others. The outcome is satisfying so I recommend publication.

Reviewer #3 (Comments to the Authors (Required)):

This resubmitted manuscript provides a significant advance in our understanding of the role of profilin in regulating microtubule nucleation from the centrosome. The authors have addressed all of my concerns and have presented a very nice study. I recommend this manuscript for publication after a few very minor changes.

1) Pfn at the centrosome in the interphase cell in figure S1A is difficult to see. Please provide an inset zooming in on the centrosome to aid the reader.

2) In the blots in figures 2 and S2, please address the bands that have been cut off in some of the boxes. I am sure these are the from IP antibody, a known background band or a band from a previous probe of the blot, but this should be indicated. For example an asterisks on the figure and a sentence in the legend.

3) On p 6. in the paragraph that starts "In our previous study we observed an indirect interaction...". I assume this experiment involved purification of the protein and testing of binding between recombinant proteins, but this is not stated. Please add this detail.

Reviewer #2 (Comments to the Authors (Required)):

The authors addressed experimentally most of the comments I raised and honestly discussed the others. The outcome is satisfying so I recommend publication.

We are pleased to learn that the referee is satisfied with our revision, and are thankful for the work and time he/she has invested by reading and commenting on our manuscripts.

Reviewer #3 (Comments to the Authors (Required)):

This resubmitted manuscript provides a significant advance in our understanding of the role of profilin in regulating microtubule nucleation from the centrosome. The authors have addressed all of my concerns and have presented a very nice study. I recommend this manuscript for publication after a few very minor changes.

We are pleased to learn that the referee now finds our revised manuscript ready for publication pending minor changes (below), and are thankful for the work and constructive criticism provided by him/her.

1) Pfn at the centrosome in the interphase cell in figure S1A is difficult to see. Please provide an inset zooming in on the centrosome to aid the reader.

Figure S1A has been modified with an inset as suggested, and to further guide the reader, the centrosome in the left-most panel (Pfn-staining) has been indicated by an arrow.

2) In the blots in figures 2 and S2, please address the bands that have been cut off in some of the boxes. I am sure these are the from IP antibody, a known background band or a band from a previous probe of the blot, but this should be indicated. For example an asterisks on the figure and a sentence in the legend.

Figures 2 and S2 have been modified as suggested and the legends altered in accordance.

3) On p 6. in the paragraph that starts "In our previous study we observed an indirect interaction...". I assume this experiment involved purification of the protein and testing of binding between recombinant proteins, but this is not stated. Please add this detail.

This sentence along with most of the rest of the text in the indicated paragraph relate to our previous study. It has now been rephrased and shifted to the Discussion section (2nd paragraph from top): "...In our paper by Nejedla et al [30] we presented data suggesting that the distribution of profilin along microtubules was dependent on formins. Since formins are well-known binding partners to profilin and have been demonstrated to attach to microtubules we concluded that the association of profilin to microtubules was at least partially indirect via formins".

The following text then indicate the experiments used in our 2016-paper to test if a direct profilin-microtubule interaction could be established: "... *in vitro* experiments involving co-sedimentation and assembly assays with purified brain tubulin and recombinant profilin did not reveal any interaction ...".

We appreciate the critical reading by the referee and hope this clarifies the issue and help the readers to follow the reasoning.

November 2, 2020

RE: Life Science Alliance Manuscript #LSA-2020-00655-TRR

Prof. Roger Karlsson
Stockholm University
Molecular Biosciences, The Wenner-Gren Institute
Svante Arrhenius väg 20C
Stockholm SE 10691
Sweden

Dear Dr. Karlsson,

Thank you for submitting your Research Article entitled "The Actin regulator Profilin 1 is Functionally Associated with the Mammalian Centrosome". It is a pleasure to let you know that your manuscript is now accepted for publication in Life Science Alliance. Congratulations on this interesting work.

DISTRIBUTION OF MATERIALS:

Again, congratulations on a very nice paper. I hope you found the review process to be constructive and are pleased with how the manuscript was handled editorially. We look forward to future exciting submissions from your lab.

Sincerely,

Shachi Bhatt, Ph.D.

Executive Editor

Life Science Alliance

<https://www.lsjournal.org/>
